

# Integrating international Chinese visualization teaching and vocational skills training: leveraging attention-connectionist temporal classification models

Yuan Yao[1], Zhujun Dai[1] and Muhammad Shahbaz[2]

[1] School of Journalism and Publishing, Jilin Engineering Normal University, Changchun, China
[2] Department of Computer Engineering, University of Engineering and Technology, Lahore, Pakistan

## ABSTRACT

The teaching of Chinese as a second language has become increasingly crucial for promoting cross-cultural exchange and mutual learning worldwide. However, traditional approaches to international Chinese language teaching have limitations that hinder their effectiveness, such as outdated teaching materials, lack of qualified instructors, and limited access to learning facilities. To overcome these challenges, it is imperative to develop intelligent and visually engaging methods for teaching international Chinese language learners. In this article, we propose leveraging speech recognition technology within artificial intelligence to create an oral assistance platform that provides visualized pinyin-formatted feedback to learners. Additionally, this system can identify accent errors and provide vocational skills training to improve learners' communication abilities. To achieve this, we propose the Attention-Connectionist Temporal Classification (CTC) model, which utilizes a specific temporal convolutional neural network to capture the location information necessary for accurate speech recognition. Our experimental results demonstrate that this model outperforms similar approaches, with significant reductions in error rates for both validation and test sets, compared with the original Attention model, Claim, Evidence, Reasoning (CER) is reduced by 0.67%. Overall, our proposed approach has significant potential for enhancing the efficiency and effectiveness of vocational skills training for international Chinese language learners.

# INTRODUCTION

As global connectivity continues to increase, the importance of international Chinese communication cannot be overstated. The allure of China's rich cultural heritage draws countless foreigners, including overseas Chinese and non-Chinese, to invest their time and energy in learning the Chinese language and experiencing its charms (*Ling, 2019*). As such, teaching Chinese as a foreign language carries significant weight in the integration and development of international Chinese language education and vocational skills training.

Corresponding author
Yuan Yao, supima123@126.com

To achieve this goal, we must adopt a new model for the diversified development of international Chinese language education that emphasizes the concept of language service and the parallel growth of careers, industries, assets, and side businesses (*Ross, Li & Gunter, 2018*). Furthermore, the integration of "Chinese language + vocational skills" must be pursued through digitalization in the age of the Internet. Traditional education methods, which rely on textbooks and teacher instruction, still hold value in modern education. However, they fail to offer the flexibility and accessibility for students to practice Chinese and evaluate their progress at any time and in any place. Thankfully, digital tools are available to support international Chinese language education. These tools use computer software to teach Chinese both online and offline and employ database technology to organize and index vast amounts of learning materials. Despite these technological advancements, the assessment of teaching quality and learning outcomes still requires the intervention of professional teachers and lacks intelligent aids.

In contemporary times, artificial intelligence (*Caiming & Yang, 2021*) is advancing rapidly and offering novel solutions for international Chinese language education and vocational skills training. Compared to traditional machine learning, deep learning holds the advantage of high learning ability and end-to-end learning (*Yann & Yoshua, 2015*). It has widespread applications in various fields such as computer vision (*Zhao et al., 2024*), natural language processing (*Yue & Zhao, 2020*), and speech recognition (*Kheddar, Hemis & Himeur, 2024*). The majority of deep learning algorithms utilize convolutional operations (*Arohan & Koustav, 2020*), and the parameters in convolution are adjustable, allowing deep learning models to adapt to various tasks autonomously by training. With the emergence of speech recognition technology, the learning and vocational skills training of international Chinese language can be enriched and empowered.

The emergence of speech recognition technology dates back to the 1950s. Initially, speech recognition involved isolated word recognition through simple template matching. In the 1960s and 1970s, RCA labs introduced a temporal regularization mechanism to address the issue of speech duration scoring (*AlJa'fari, 2021*). From the 1980s onward, statistical model-based speech recognition methods were developed, with acoustic models mainly based on Markov models (hidden Markov model, HMM) (*Deng & Söffker, 2021*). However, traditional HMM-based models encounter several challenges. For one, the training process of HMM-based models is intricate and challenging to optimize. Additionally, traditional HMM-based speech recognition models often have disparate training data and optimization criteria for different modules. The primary problem we aim to address is how to assist international Chinese language teaching through deep learning speech recognition methods.

Despite significant advancements in the field of international Chinese language teaching, there remains a notable gap in the effective application of deep learning methods. Traditional approaches often rely on outdated teaching materials and methods that do not leverage the latest technological innovations. This leads to inefficiencies in teaching and learning processes. Furthermore, the shortage of qualified instructors who can integrate deep learning techniques into language education exacerbates these challenges. Existing digital tools for language learning are underutilized and lack the sophistication needed to

provide personalized, real-time feedback and assessment. The current methods do not adequately incorporate advanced speech recognition and natural language processing technologies, limiting their ability to accurately assess and correct pronunciation and accent errors.

The motivation for this study arises from the need to innovate within the realm of international Chinese language teaching by employing state-of-the-art deep learning techniques. Our goal is to address the limitations of traditional methods by developing a more intelligent, efficient, and accessible approach to language education. Specifically, we aim to enhance the learning experience through the use of advanced speech recognition technology that provides real-time, visualized pinyin-formatted feedback to learners. Additionally, our system is designed to identify and correct accent errors, a feature that is not commonly found in existing language learning tools. By integrating vocational skills training with language education, we seek to offer a holistic learning experience that prepares learners for practical, real-world applications. The use of the Attention-CTC model, which incorporates a temporal convolutional neural network, represents a significant innovation in capturing location information for accurate speech recognition.

## RELATED WORKS

Traditional speech recognition models, such as GMM/DNN-HMM (*Jian & Xuan, 2020*), are complex and require extensive expertise to develop. In contrast, end-to-end models have gained traction in recent years for their relative simplicity. Unlike traditional hybrid models, end-to-end models can directly map speech sequences to output labels. Since the process of building an end-to-end speech recognition system requires less domain-specific knowledge, the process of model building and training is simplified, which lowers the threshold for entry into speech recognition. Theoretical assumptions inherent in traditional speech recognition techniques are being addressed by the emergence of end-to-end models, making them a hot topic of research.

Around 2006, *Graves et al. (2006)* proposed a completely end-to-end training method for speech recognition, called Connectionist Temporal Classification (CTC). Unlike traditional methods, CTC does not require pre-alignment operations and can achieve sequence-level alignment of input speech and output text during training. Experiments on the TIMIT speech *corpus* demonstrated the superiority of CTC over baseline HMM and hybrid HMM-RNN (*Liang & Rong, 2016*), achieving a label error rate (LER) of 30.51%. This development opened the door for the exploration of fully end-to-end speech recognition algorithms. In 2015, Baidu proposed the Deep Speech system (*Awni & Carl, 2014*; *Amodei et al., 2016*), which was built based on CTC training methods and achieved state-of-the-art results on the Switchboard dataset at that time, leading to its commercial implementation. Since then, RNN and attention-based codec structure methods (*Shinji & Takaaki, 2017*), RNN-T methods (*Wang et al., 2023*), and CTC modeling methods (*Fan et al., 2023*) have been successfully applied to various speech recognition tasks, producing results that are comparable to those of traditional speech models. Despite the comparable accuracy of end-to-end models to traditional hybrid models in ASR, traditional hybrid models are still predominantly used in most commercial ASR systems. This is because ASR

accuracy is not the only factor that determines the production choice between hybrid and end-to-end models.

The introduction of the Transformer model has fundamentally reshaped the landscape of natural language processing (NLP), establishing a robust foundation for numerous subsequent advancements. The Transformer model, characterized by its attention mechanisms, enables efficient parallelization during training, addressing limitations inherent in recurrent neural networks (RNNs) and long short-term memory (LSTM) networks. This paradigm shift led to significant enhancements in various NLP tasks, spurring a wave of innovative models.

Among these, the Transformer-XL model by *Dai et al. (2019)* represents a notable improvement. By introducing segment-level recurrence and a new positional encoding scheme, Transformer-XL overcomes the fixed context length limitation of standard Transformers, thereby enabling better handling of longer sequences. This model's success in enhancing language modeling and other NLP tasks underscores the versatility and scalability of the Transformer architecture. Similarly, the BERT model, which employs bidirectional Transformers (*Reza et al., 2023*), revolutionized NLP by enabling deeper contextual understanding. BERT's masked language model and next sentence prediction objectives facilitate pre-training on large corpora, yielding substantial gains across a wide array of NLP benchmarks. The ability to fine-tune BERT for specific tasks further underscores its adaptability and has set a new standard for pre-trained language models.

Extending the influence of Transformers beyond NLP, *Linhao & Shuang (2018)* adapted the Transformer model to the speech recognition domain, resulting in the Speech-Transformer model. By leveraging the self-attention mechanism, this model improved accuracy and reduced training time compared to traditional RNN-based speech recognition models. This adaptation demonstrates the Transformer architecture's potential in handling diverse data types and tasks. In 2020, the Conformer model emerged, combining convolutional neural networks (CNNs) with Transformers to enhance speech recognition performance. The Conformer model's integration of convolutional operations addresses the locality issue of Transformers, thereby capturing both local and global dependencies more effectively. This convolution-enhanced architecture has further advanced the state-of-the-art in speech recognition, highlighting the ongoing evolution and refinement of Transformer-based models.

Despite these significant advancements, Transformer-based models exhibit notable limitations, particularly in streaming speech recognition. Modifying the attention mechanism to support streaming recognition often results in high latency and increased error rates (*Rolland & Abad, 2024*). These challenges underscore a critical gap in current research, as real-time applications necessitate efficient and accurate streaming capabilities.

Our current challenge is to develop solutions that address these limitations, with a specific focus on creating an international Chinese-assisted teaching system that utilizes advanced speech recognition methods. By critically evaluating and building upon existing Transformer-based models, we aim to overcome the barriers of high latency and error rates in streaming speech recognition, thereby enhancing the effectiveness and applicability of these models in real-world educational settings. This endeavor not only

seeks to refine technical performance but also aims to broaden the accessibility and utility of speech recognition technologies across diverse linguistic and cultural contexts.

# INTERNATIONAL CHINESE TEACHING AID SYSTEM

In order to meet the demand for visual instruction in international Chinese and improve the efficiency of vocational skills training, this article proposes an innovative teaching aid system based on pinyin modeling and deep learning. The system's purpose is to automatically identify Chinese speech spoken by international Chinese learners and provide accurate pinyin and pronunciation based on the meaning of the input speech sequence. Figure 1 illustrates the system's overall process flow.

The international Chinese teaching aid system, depicted in the figure, is comprised of three components: the input layer, the acoustic model layer, and the output layer. The input layer performs speech segmentation, pre-processing, and feature sequence formation. The Mel-scale Frequency Cepstral Coefficients (MFCC) method is utilized for feature extraction, along with windowing and fast Fourier transform, as detailed in "Feature modeling". The acoustic model layer is the central component of speech recognition, and incorporates the attention mechanism, CTC model, and deep convolutional neural network. Lastly, the output layer generates the recognized Chinese pinyin, including both the pinyin body and vocal tones.

## Feature modeling

In order to enable the computer to process speech data, it is necessary to extract features from the input speech and transform it into a format that can be recognized by the computer. Common feature extraction methods include FilterBank (FBank), Mel-scale Frequency Cepstral Coefficients (MFCC), Discrete Wavelet Transform (DWT), and so on. In this article, we have chosen the MFCC feature extraction method, which has more DCT cepstral coefficients compared to the FBank method. This helps remove correlations between signals in each dimension and map signals to a low-dimensional space. The detailed steps of the feature extraction process are illustrated in Fig. 2.

As depicted in Fig. 2, initially, the input acoustic signal undergoes pre-emphasis in order to amplify the higher frequencies. The objective of this process is to equalize the spectrum, thereby preventing numerical issues during the Fourier transform operation, as well as to enhance both the signal and the signal-to-noise ratio. The equation for the pre-emphasis filter applied to the input signal can be calculated in Eq. (1):

$$y(t) = x(t) - \alpha x(t-1) \tag{1}$$

where $y$ is the output signal, $t$ is the time, and $\alpha$ is the filter coefficient, which in this article is 0.95. The subsequent phase involves segmenting the signal into frames. Due to the time-varying nature of the signal's frequency, it is generally unproductive to Fourier transform the entire signal. In this article, a short frame of 25 milliseconds is Fourier transformed. Following the Fourier transform, a triangular filter is utilized to extract the frequency bands from the power spectrum. In this article, a total of 40 filters are selected, and a filter_banks plot is obtained after this step. Next, discrete cosine transform (DCT) is
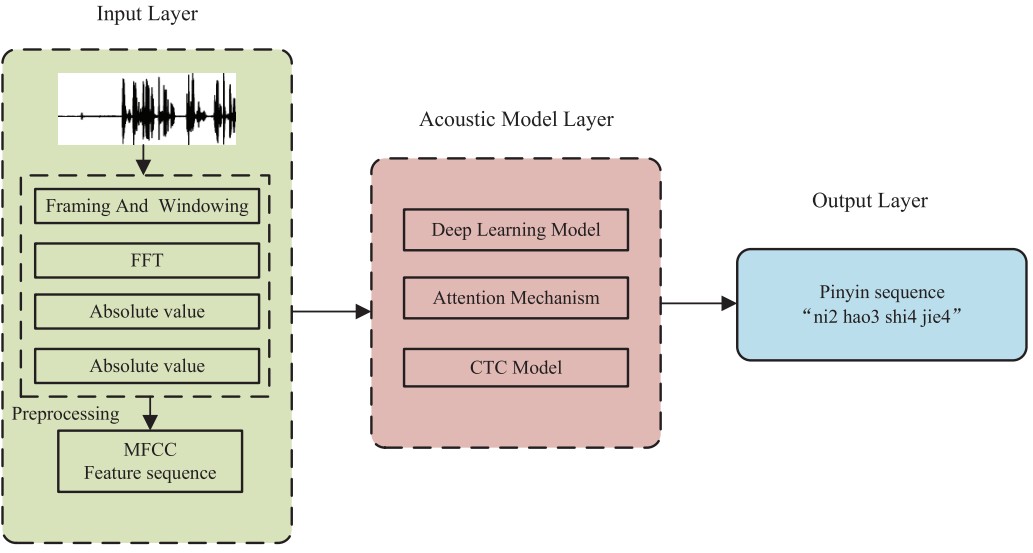

**Figure 1  Structure of an international Chinese teaching aid system.**

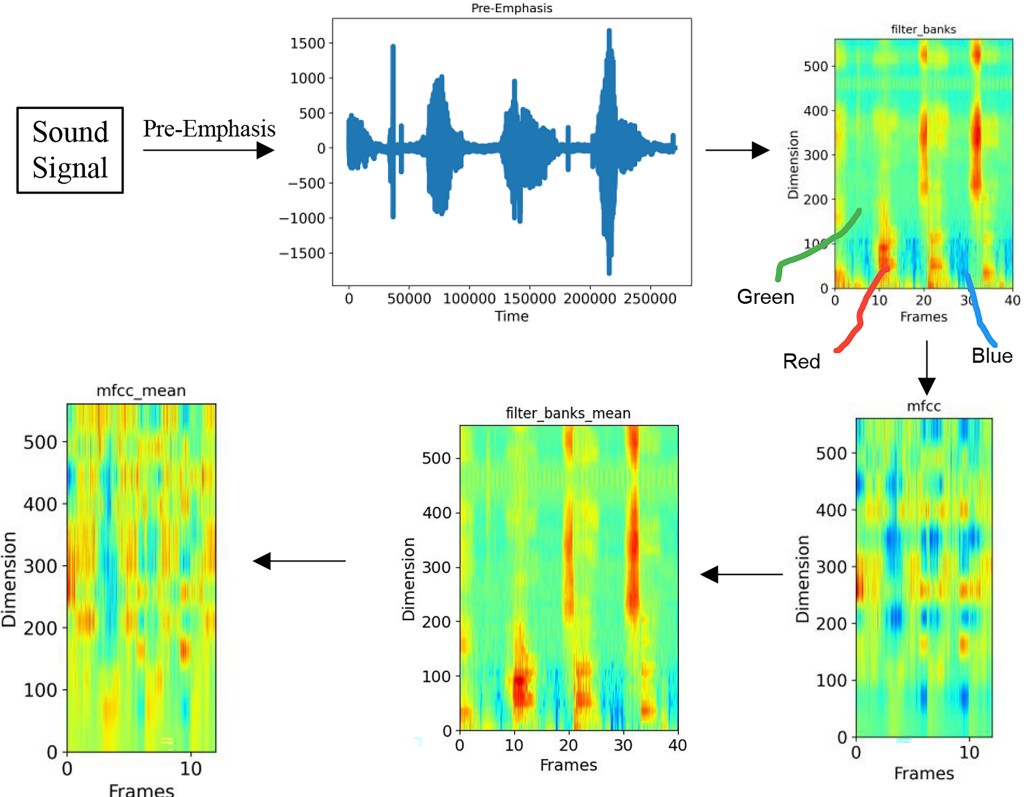

**Figure 2  Schematic diagram of feature modeling process.**

employed to decorrelate the filter bank coefficients and create a compressed representation of the filter bank. For automatic speech recognition (ASR) purposes, this article retains cepstrum coefficients 2–13, while discarding the remainder, as they signify rapid changes in the filter bank coefficients that do not aid in ASR. The fourth stage involves balancing the spectrum and improving the signal-to-noise ratio (SNR) by subtracting the average value of each coefficient from all frames, thereby resulting in the final feature modeling outcomes. All of the aforementioned procedures are designed to render the time-domain sound signal discernible by the computer during the speech recognition process, and to enhance the visibility of the features during the recognition process.

## Attention-CTC speech recognition model

The configuration of the Attention-CTC speech recognition model put forth in this article is depicted in Fig. 3. It is comprised of a pre-processing module (consisting of an acoustic pre-module and a text pre-module), an encoder-decoder, and a hybrid CTC/attention loss. The model regards speech recognition as a sequence-to-sequence mission, with the encoder mapping the input frame-level acoustic characteristics to a high-level sequence representation, and the decoder generating results by jointly decoding the already produced text with attention-adjusted hidden states. Ultimately, the decoder produces the intended transcribed sequence.

The structure of the preprocessing module is shown in Fig. 4, which is divided into an acoustic preprocessing module and a text preprocessing module. In the acoustic pre-processing module, $K$ convolutional blocks (2-D) are used, and each convolutional block contains one 2-D convolutional layer and one ReLU activation layer. Finally, positional encoding is used to obtain the absolute position information of the acoustic features, and the structure is shown in Fig. 4A. In the text front module, $J$ TCN modules are used to learn the implied location relationships, and the specific structure is shown in Fig. 4B. Speech recognition can be interpreted as a timing-related predicament, and the temporal convolutional network (TCN) architecture utilized in this article outperforms the conventional convolutional structure in addressing this issue. Unlike the recurrent neural network (RNN) architecture, the causal convolution employed in TCN imposes a robust timing constraint, while the expansion convolution can be adjusted through the expansion factor to improve the perceptual field. This article leverages the TCN architecture in the previous model to maintain the original parallelism of Attention while ensuring greater gradient stability.

Figure 5 illustrates the architecture of the encoder and decoder, which comprises several identical modules stacked together, with each module including two main sub-layers: the multi-head attention layer and the feed forward network layer (feed forward). Each sub-layer uses residual connection and layer normalization. The decoder varies from the encoder in that it adopts a multi-head attention mechanism that masks future information to ensure that future label information is not accessible during decoding. Additionally, the second multi-head attention layer utilizes cross-attention. In this article, the encoder-decoder structure is adjusted to integrate a parallel TCN structure into the encoder section, which collaborates with the multi-head attention layer to extract additional features while

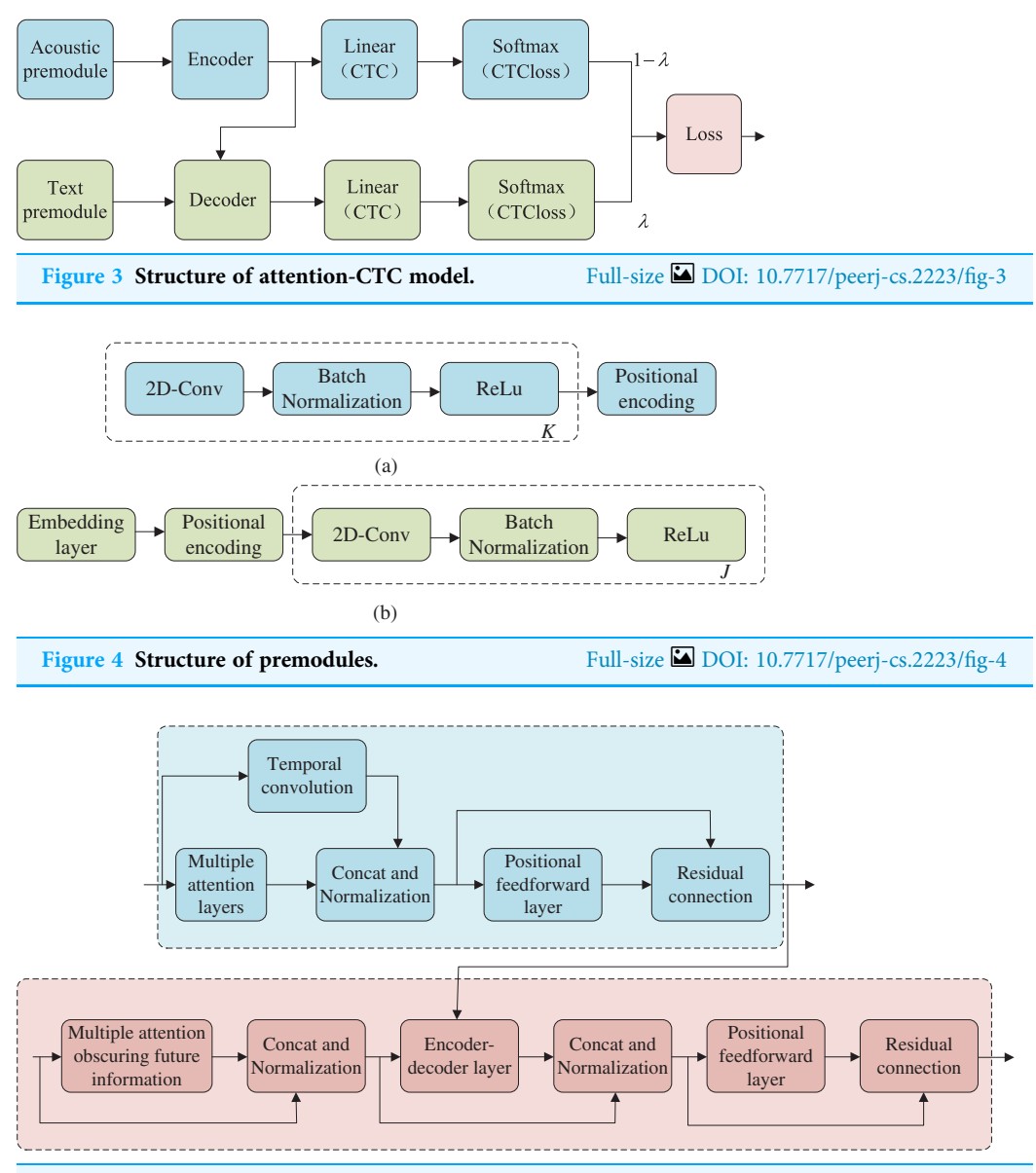

**Figure 3** Structure of attention-CTC model.

**Figure 4** Structure of premodules.

**Figure 5** Structure of encoder-decoder.

slowing the degradation of location information. Furthermore, the encoder output component is also fed into the CTC structure to accelerate model training convergence and improve the model's robustness.

The CTC objective function is often utilized as an auxiliary task in numerous modeling endeavors, and can boost the model's performance. This is because, unlike the attention model, the CTC forward-backward algorithm enables forced monotonic alignment between speech and tag sequences, compensating for the absence of the attention alignment mechanism and enhancing the model's robustness in noisy external environments. The tcn-Transformer-CTC model leverages the benefits of both CTC and attention, and the total loss function is defined as a weighted sum of the CTC loss and attention loss, as shown in Eq. (2):

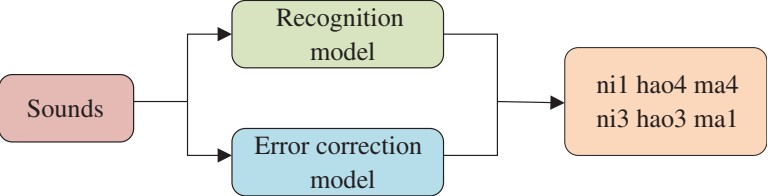

**Figure 6 Flow of recognition and error correction model.**

$$T_{loss} = \lambda CTC_{loss} + (1 - \lambda) ATT_{loss}. \qquad (2)$$

### Identification and error correction model

To adapt the approach proposed in this article to the requirements of international Chinese vocational skills training, two datasets were used to train the model to output both the pinyin spoken by the participants and the correct pinyin identified by the semantics. The first dataset was a public dataset of standard Chinese, while the second dataset was a fuzzy Chinese dataset created specifically for this purpose. The recognition process of the international Chinese language education visualization system is illustrated in Fig. 6.

The Chinese datasets employed in this study were obtained from multiple publicly available datasets, which are elaborated upon in the subsequent section. The fuzzy Chinese dataset that was constructed in this article, on the other hand, was compiled from web-based recordings of individuals learning Chinese. This includes recordings from television programs, contact sessions, and Chinese speech recordings. The self-created dataset is composed of 123 h of data collected from 254 Chinese learners hailing from different countries.

## EXPERIMENT AND ANALYSIS

The proposed technique underwent training and testing on four Chinese corpora that are publicly available, namely THCHS-30 (DOI 10.5281/zenodo.10210310), AISHELL-1 (DOI 10.5281/zenodo.10964183), MAGICDATA (DOI 10.5281/zenodo.10964199), and Aidatatang_200zh (DOI 10.5281/zenodo.10964199). The four corpora cover data from different domains and language styles. THCHS-30 contains Chinese speech data from different speakers, AISHELL-1 focuses on Mandarin speech recognition, MAGICDATA covers Chinese speech in multiple application scenarios, and Aidatatang_200zh contains Chinese speech data with multiple accents and backgrounds. By training and testing on these corpora, the generalization ability and adaptability of the model on different speech samples can be evaluated.

THCHS-30, curated and released by the Center for Speech and Language Technology (CSLT) at Tsinghua University, is comprised of 1,000 news passages that have been divided into four sets of 250 sentences each, read aloud by ten individuals. AISHELL-1, an open-source speech dataset published by Beijing Hill Shell, contains roughly 178 h of speech data. Aidatatang_200zh mainly consisting of mobile recordings, and involved the

participation of 600 speakers with diverse accents from different regions of China. MAGICDATA, provided by Beijing AIDU Intelligence Technology Co., Ltd., consists of about 755 h of speech data obtained from 1,080 Mandarin speakers originating from various accent regions in mainland China. The training set, validation set, and test set were allocated in a 51:1:2 ratio. In addition, the proposed model for recognizing fuzzy Chinese speech by international Chinese learners will be trained and tested independently on the dataset developed in this study. The experimental hardware/software environment is shown in Table 1.

## Evaluation indicators

The objective of a speech recognition model is to forecast the correct textual outcome that corresponds to the given input speech with utmost accuracy. As users of speech recognition systems are highly susceptible to errors in the results, the performance of a Mandarin speech recognition model is commonly assessed by means of the character error rate (CER), which typically varies between 0% to 100%. The CER value, which is essentially a measure of accuracy, decreases with the precision of the recognized speech, implying improved model performance. In this chapter, we evaluate the CER value at the pinyin level, which is determined by computing the edit distance, *i.e.*, the minimal number of modifications needed to transform the predicted pinyin sequence to the correct pinyin label sequence. The formula for CER is given by Eq. (3).

$$CER = \frac{S + D + I}{N} \tag{3}$$

where $N$ is the length of the correct tag sequence, $S$ is the number of replaced characters, $D$ is the number of missed characters, and $I$ is the number of additional characters added. The average CER on the test set is generally used to represent the model performance.

## Ablation experiment

To substantiate the advantages of the proposed method, we have devised ablation experiments to ascertain the efficacy of combining CTC models with TCN structures. The experiments have been conducted in two parts, one involving standard Chinese recognition, which involves training models on the four public datasets chosen for this study, and the other involves fuzzy Chinese recognition, which involves training models on the self-constructed datasets. The training strategies have been designed as follows: (1) The original Attention model. (2) Attention+TCN, which entails the incorporation of the TCN structure outlined in this study into the original Attention model. (3) Attention+CTC, which includes the TCT module in the original Attention model. (4) Attention+TCN +CTC, the approach proposed in this article. The training strategies employed for all models, along with their hyperparameter settings, are summarized in Table 2. The results for standard Chinese recognition are presented in Table 3, and those for fuzzy Chinese recognition are reported in Table 4.

Table 2 exhibits the hyperparameters employed in the training process. All models have been trained for 250 epochs to guarantee convergence. The batch size has been set to 64,

**Table 1 Experimental environment.**

| Category | Details |
|---|---|
| Processor | Intel(R) Xeon(R) CPU E5-2620 @ 2.40 GHz |
| Running memory | 64 GB |
| GPU | NVIDIA tesla V100 |
| Operating system | Ubuntu 20.0.4 |
| Software | Pytorch 1.11.0 |

**Table 2 Hyperparameter setting.**

| Parameter | Setting |
|---|---|
| Epochs | 250 |
| Batch size | 64 |
| Optimizer | Adam |
| Learning rate | 0.001 |
| Dropout rate | 0.1 |

**Table 3 Standard Chinese recognition results.**

| Methods | Val set CER (%) | Test set CER (%) |
|---|---|---|
| Attention | 6.54 | 6.72 |
| Attention+TCN | 6.32 | 6.43 |
| Attention+CTC | 6.01 | 6.14 |
| Ours | 5.87 | 5.98 |

**Table 4 Fuzzy Chinese recognition results.**

| Methods | Val set CER (%) | Test set CER (%) |
|---|---|---|
| Attention | 5.98 | 6.12 |
| Attention+TCN | 5.78 | 5.93 |
| Attention+CTC | 5.67 | 5.89 |
| Ours | 5.50 | 5.63 |

and the optimizer has been selected from the Adam optimization algorithm. The learning rate has been set to 0.001, and the dropout ratio is 0.1. Table 3 illustrates the outcomes of this study's method in standard Chinese recognition, and it is evident that the proposed modifications, in conjunction with TCN and CTC, have led to a noticeable improvement in the speech recognition performance of the model. The combined experimental results of the proposed method on the four datasets reveal a 0.67% reduction in CER on the validation set and a 0.74% reduction in CER on the test set when compared to the original

**Table 5 Experimental results of standard Chinese recognition comparison.**

| Methods | Test set CER (%) |
| --- | --- |
| Chain | 7.42 |
| Nnet | 7.31 |
| Espnet | 6.54 |
| CTC/Attention | 6.12 |
| Ours | 5.98 |

**Table 6 Comparative experiment results of fuzzy Chinese recognition.**

| Methods | Test set CER (%) |
| --- | --- |
| Chain | 7.14 |
| Nnet | 7.08 |
| Espnet | 6.31 |
| CTC/Attention | 5.98 |
| Ours | 5.63 |

Attention model. Table 4 presents the results of the ablation experiments conducted on the self-constructed fuzzy recognition dataset. These results demonstrate that the CER of this method has been reduced by 0.48% on the validation set and by 0.49% on the test set. The findings of this study affirm that the proposed method can effectively reduce the error rate in the Chinese recognition process.

## Model comparison

This article has compared our proposed method with several similar methods to further illustrate its advantages. These acoustic models include traditional speech recognition methods such as chain and nnet, Transformer in ESPnet, and CTC/Attention. The comparative results on both datasets are presented in Tables 5 and 6.

This study compares several acoustic models to evaluate the efficacy of a novel approach in Chinese speech recognition. The experimental results, as shown in Tables 5 and 6, illustrate the performance of traditional methods like chain and nnet, alongside modern frameworks such as ESPnet with Transformer and CTC/Attention architectures. Across both standard and fuzzy Chinese recognition datasets, our proposed model consistently outperformed all others in terms of CER. Specifically, in Table 5, our model achieved a CER of 5.98%, surpassing the closest competitor, CTC/Attention, by a margin of 0.14 percentage points. Similarly, in Table 5, our model demonstrated a CER of 5.63%, again outperforming CTC/Attention, which achieved 5.98%.

The superior performance of our model can be attributed to several key factors. Firstly, our model architecture incorporates advanced attention mechanisms that efficiently capture contextual dependencies in speech sequences, thereby enhancing both phonetic and semantic alignment. This innovation in attention modeling is crucial in improving
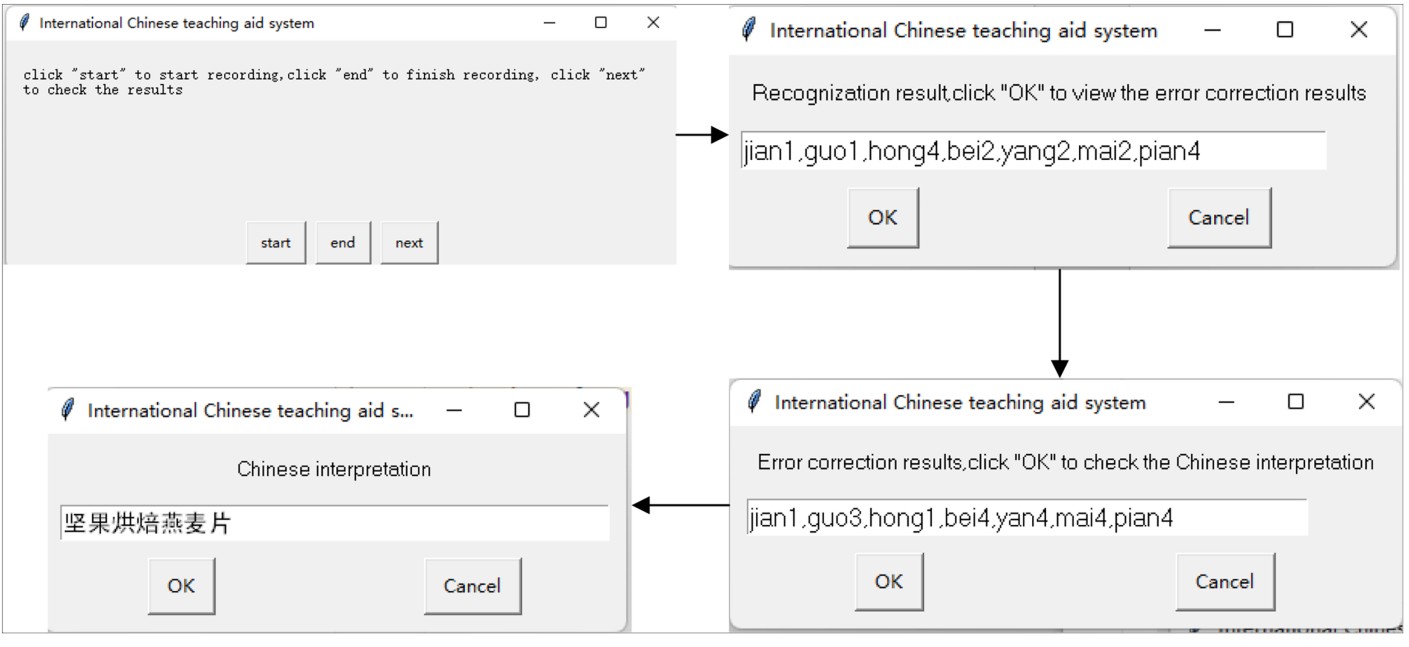

**Figure 7  Visualization results.**                                                 

accuracy, particularly in scenarios involving ambiguous or variable speech patterns. Secondly, our training strategy leverages extensive data augmentation techniques and optimized preprocessing steps, which mitigate noise and variability in speech data, contributing to robust performance across different recognition tasks. Additionally, our model benefits from computational optimizations that enhance efficiency without compromising accuracy, making it well-suited for real-time applications where speed and reliability are paramount.

These findings suggest significant advancements in Chinese speech recognition technology, highlighting the effectiveness of our approach in achieving lower CERs compared to established methods. Beyond the experimental results presented here, future research could explore additional metrics to further validate the practical applicability of our model in diverse real-world settings. By continuing to refine attention mechanisms and expanding training datasets, further improvements in accuracy and robustness can be anticipated, paving the way for enhanced speech recognition systems in various domains of computational linguistics and artificial intelligence.

## Visualization results

In Fig. 7 presents selected recognition results to illustrate the process of the proposed international Chinese visual teaching aid system, including the recognition and error correction processes. The system includes a real-time recording interface, and the results of the recording are recognized by the two models to show the recognition results and the correct results, respectively. This visual learning tool can serve as an effective aid for

students' vocational skills training, enabling them to integrate their international Chinese speaking training into their daily life and practice anywhere and anytime.

## CONCLUSION

In this manuscript, we present a novel speech recognition algorithm specifically designed for international Chinese teaching and vocational skills training. Our proposed algorithm integrates the TCN architecture, an Attention mechanism, and CTC model to achieve superior accuracy in Chinese speech recognition. This approach not only minimizes error rates but also enhances the robustness of recognizing varied speech patterns inherent in educational and vocational contexts. A key innovation of our work is the incorporation of a fuzzy Chinese recognition function. This function identifies and corrects pronunciation errors of Chinese learners by leveraging both standard Chinese pronunciation models and fuzzy Chinese models, which accommodate variations in pronunciation commonly encountered in educational settings. By facilitating more accurate and adaptive learning experiences, our system contributes significantly to the acquisition and mastery of spoken Chinese, thereby promoting the integration of Chinese language education with vocational training initiatives.

In our future research endeavors, we aim to expand the scope and diversity of our datasets. This includes augmenting our training dataset with a broader range of speech samples encompassing various accents and dialects found across China. Additionally, we plan to enhance our fuzzy dataset to better capture and correct pronunciation variations. Furthermore, integrating a comprehensive *corpus* of diverse Chinese dialects and national accents will enable our system to adapt more effectively to regional linguistic nuances. Our ultimate aspiration is to develop a versatile system that not only improves Chinese language learning outcomes but also incorporates practical functionalities tailored to the specific needs of learners in different educational and vocational contexts. This includes enhancing adaptability to regional dialects, integrating multimodal data inputs for more comprehensive learning experiences, and exploring advanced techniques to further optimize speech recognition performance in educational settings.

## ACKNOWLEDGEMENTS

We thank the anonymous reviewers whose comments and suggestions helped to improve the manuscript.

### Funding

This work was supported by the subordinate organization of the China Vocational and Technical Education Society in 2023 "Research on the Innovation Model and Practice of Integrated Publishing for Vocational Education from the Perspective of Sustainable Development" (ZJ2023B178); the Education and Teaching Research Project of Jilin Engineering Normal University "Research on the optimization of the curriculum system of editing and publishing major in local undergraduate universities based on the needs of the

industry" JGSZ[2022] No. 170 (8); The Second Planning Project of Huang Yanpei's Vocational Education Thought Research "Innovation and Practice of the 'Government-Industry-School' Collaborative Education Mechanism under the Perspective of Huang Yanpei's Comprehensive Vocational Education Thought" (ZJS2024YB222); the University level doctoral project of Jilin Engineering Normal University "Research on the transformation and development trend of China's book publishing industry under the background of media convergence" (BSSK201906). The funders had no role in study design, data collection and analysis, decision to publish, or preparation of the manuscript.

## Grant Disclosures

The following grant information was disclosed by the authors:

Subordinate organization of the China Vocational and Technical Education Society: ZJ2023B178.

Education and Teaching Research Project of Jilin Engineering Normal University: JGSZ [2022] No. 170 (8).

Huang Yanpei's Vocational Education Thought Research: ZJS2024YB222.

Jilin Engineering Normal University: BSSK201906.

## Competing Interests

The authors declare that they have no competing interests.

## Author Contributions

- Yuan Yao conceived and designed the experiments, performed the experiments, analyzed the data, prepared figures and/or tables, and approved the final draft.
- Zhujun Dai conceived and designed the experiments, performed the experiments, performed the computation work, authored or reviewed drafts of the article, and approved the final draft.
- Muhammad Shahbaz conceived and designed the experiments, performed the experiments, analyzed the data, authored or reviewed drafts of the article, and approved the final draft.

## Data Availability

The code is available in the Supplemental File.

The THCHS-30 dataset is available at Zenodo: Taubert, S. (2023). THCHS-30 Chinese TTS (0.0.1). Zenodo. https://doi.org/10.5281/zenodo.10210310.

The AISHELL-1 dataset is available at Zenodo: None. (2024). AISHELL-1 [Data set]. Zenodo. https://doi.org/10.5281/zenodo.10964183.

The MAGICDATA dataset is available at Zenodo: None. (2024). MAGICDATA [Data set]. Zenodo. https://doi.org/10.5281/zenodo.10964199.

The Aidatatang_200zh dataset is available at Zenodo: None. (2024). Aidatatang_200zh [Data set]. Zenodo. https://doi.org/10.5281/zenodo.10964211.

## Supplemental Information

Supplemental information for this article can be found online at http://dx.doi.org/10.7717/peerj-cs.2223#supplemental-information.

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
