# Peer review of "Integrating international Chinese visualization teaching and vocational skills training: leveraging attention-connectionist temporal classification models"

_PeerJ Computer Science, doi:10.7717/peerj-cs.2223_

## Round 0.1 · original submission · Major Revisions

Dear authors,

Thank you for submitting your article. Feedback from the reviewers is now available. We strongly recommend that you address the issues raised by the reviewers, especially those related to readability, experimental design and validity, and resubmit your paper after making the necessary changes. Before submitting the paper following should also be addressed:

1. Please write research gap and the motivation of the study. Evaluate how your study is different from others. More recent literature should be examined.
2. The values for the parameters of the algorithms selected for comparison are not given.
3. The paper lacks the running environment, including software and hardware. The analysis and configurations of experiments should be presented in detail for reproducibility. It is convenient for other researchers to redo your experiments and this makes your work easy acceptance. A table with parameter settings for experimental results and analysis should be included in order to clearly describe them.
4. Equations should be used with equation number. Please do not use “following”, “as follows”, etc. Explanation of the equations should be checked. All variables should be written in italic as in the equations. Their definitions and boundaries should be explained. Relevant references should be given for the equations.

Best wishes,

Reviewer 1 ·

Basic reporting

The language used in the manuscript is simple clear, business-like English, and the manuscript is not badly structured and complies with the journal’s requirements. The context established in the introduction is sufficient, and it explains why improving the practice of teaching the Chinese language is essential to using superior AI methods. The literature review is quite comprehensive, identifies previous research in the field, and explains how the specific study addresses a gap in the existing knowledge. Including the figures and tables is appropriate and useful; the legends are clear and concise and help make the report easy to understand. Raw data has been stated as supplied, permitted by the PeerJ policy.

Recommendations for Revision:

Perhaps it is possible to elaborate on the background a bit more by referencing more recent research that would support the case for the need for the proposed AI model given the current trends in technology advancements in the learning environment.

Check that all sources are correct and formatted properly. In some cases, details could be needed to clarify the explanation of how the reference was used.

Experimental design

The research question is concise, accurate, and significant in its attempt to respond to a particular absence in the combination of AI with vocational training in language learning. The methodological design seems quite well done, with a comprehensive explanation of the system design, data processing and feature extraction, which should make it easy to replicate the study. In terms of ethical considerations, they are complied with, and there do not seem to be any weaknesses in the conduct of experiments.

Recommendations for Revision:
More details regarding the criteria concerning the selection of datasets could be included in the methods section and more elaboration on why the Attention-CTC was deemed appropriate.

It might also have been beneficial to focus more on the limitations of the chosen methods to recognise possible sources of bias or points of improvement.

Validity of the findings

The conclusions drawn are compelling, and the statistical analysis carried out seems to be quite valid. The conclusions are presented concisely, are in line with the objectives outlined in the study, and are based on the research findings. Commendable efforts are made to provide the implications of the work done in the larger context of language education and Artificial Intelligence.
Recommendations for Revision:

Extend the statistical analysis to compare the proposed model's relative performance with others in terms of efficiency, scalability, and genuine usability.

Indeed, to increase the scope of the research, one might consider applying the model to other languages or linguistic environments.

Additional comments

All in all, this work contains a well-thought-out idea that can provide useful findings for AI applications in language education. The topic of integrating AI to support vocational education and training in Chinese is timely and relevant. I found the paper well-prepared and organised, but it had some mistakes that, if addressed, would enhance its quality.

Recommendations for Revision:
The authors should also provide more information on how readers can experience the proposed system. This can be in supplementary material or a link where readers can get a demo of the proposed system. This would be beneficial in making the physical comprehension of the system more profound.

It may also be necessary to incorporate information from real users of the system with their quantitative results to offer qualitative insights about the functionality and usability of the system.

I have no complaints about the paper: it is coherent and presents a timely and methodologically sound investigation. In toto, some minor changes, specifically more development of some sections, can allow the paper to positively contribute to the discussion on the use of AI in education.

Annotated reviews are not available for download in order to protect the identity of reviewers who chose to remain anonymous.

Reviewer 2 ·

Basic reporting

I have carefully reviewed your manuscript titled "Application of Artificial Intelligence in the Integration of International Chinese Visualization Teaching and Vocational Skills Training" and commend your innovative approach. Below are my detailed comments and suggestions from a technical perspective:

Title: Starting with the title, it is informative but could be more specific. Consider adding terms like "using Attention-CTC Models" to highlight the technical focus of the paper.

Abstract: The abstract should provide more technical details, such as the key architectural elements of the Attention-CTC model and specific performance metrics achieved.

Introduction (Paragraph 1): Moving to the introduction, it should clearly define the research gap. Include a discussion on the limitations of existing AI methods in educational technology and how your approach addresses these gaps.

Related Works (Section 2): The related works section needs a more detailed critique of the existing literature. Highlight the strengths and weaknesses of related works and position your research within this context.

Experimental design

Methodology (Section 3): In terms of methodology, the technical description of the Attention-CTC model should include diagrams of the network architecture. Provide detailed descriptions of each layer and the flow of data through the network.

Experimental Design (Section 4): Regarding the experimental design, detail the experimental protocol, including the hardware and software environment used for training and testing. Discuss the choice of optimization algorithms and learning rates.

Validity of the findings

Results (Section 4.3): The results section should include more comprehensive tables and figures that illustrate the performance of your model compared to baseline methods. Use precision, recall, and F1-score to provide a fuller picture of performance.

Discussion (Section 5): The discussion should critically analyze the results, explaining why your model outperforms others. Include insights into the potential reasons behind any observed performance improvements or degradations.

Conclusion: The conclusion should summarize the technical contributions of your work and suggest concrete future research directions, such as enhancing the model's adaptability to different dialects or integrating multimodal data.

Additional comments

References: Finally, ensure all references are up-to-date and relevant. Include seminal papers on AI in education and recent advancements in attention mechanisms and CTC models.

Your research presents an innovative application of AI in educational technology. Addressing these technical comments will enhance the clarity and impact of your manuscript.

---

## Round 0.2 · accepted · Accept

Dear authors,

Thank you for clearly addressing all the reviewers' comments. The paper now seems to be ready for publication in light of this revision.

Best wishes,

Reviewer 1 ·

Basic reporting

The manuscript is professionally written in English. It has a thorough introduction, well-referenced literature, high-quality figures, and a structure conforming to PeerJ standards, ensuring easy understanding and navigation.

Experimental design

The research addresses a significant need in Chinese language education using rigorous, ethical methodologies. It has been revised with detailed descriptions of the running environment, software, hardware, and parameter settings for better reproducibility.

Validity of the findings

The manuscript provides robust, statistically sound findings, with detailed parameter values for comparison algorithms, well-formatted equations, and logical conclusions, all linked to the original research question.

Additional comments

The manuscript has been enhanced with detailed discussions on limitations and future work and expanded explanations of the experimental setup, enhancing its quality and relevance. The manuscript is recommended for acceptance due to its innovative and practical AI contribution to language education, providing a solid foundation for future research and educational applications.

Reviewer 2 ·

Basic reporting

Overall update of the article is quite reasonable

Experimental design

Comments regarding experimental design are well addressed

Validity of the findings

Paper is in acceptable form